The mycorrhizal fungi of Cymbidium promote the growth of Dendrobiumofficinale by increasing environmental stress tolerance

Li Yulong 1
Kang Zhihua 2
Zhang Xia 3
Sun Ping 4
Jiang Xiaohui 5
Han Zhengmin zmhan_njfu@163.com 1
1 College of Forestry, Nanjing Forestry University , Nanjing , China
2 Jiangsu Academy of Agricultural Sciences , Nanjing , China
3 Suqian Forest Pest Quarantine Control Station , Suqian , China
4 Jiangsu Aosaikang Pharmaceutical Co., Ltd , Suzhou , China
5 Garden Bureau, Management Committee of Huangshan Scenic Area , Huangshan , China
Kaminaka Hironori
Electronic publication date: 2021 Dec 6
Publication date: 2021
Volume: 9
Electronic Location ID: e12555
Received 2021 Mar 30; Accepted 2021 Nov 5
Copyright: ©2021 Li et al.
Copyright year: 2021
Copyright holder: Li et al.
License: This is an open access article distributed under the terms of the Creative Commons Attribution License, which permits unrestricted use, distribution, reproduction and adaptation in any medium and for any purpose provided that it is properly attributed. For attribution, the original author(s), title, publication source (PeerJ) and either DOI or URL of the article must be cited.
License URL: https://creativecommons.org/licenses/by/4.0/

Keywords: Mycorrhizal fungus, Root rot, Dendrobium officinale, Drought tolerance, Biocontrol

Funding: Science and Technology Project of Huangshan City 2017KN-06 Nanjing Forestry University Research and Development project JZ20190011 Priority Academic Program Development of Jiangsu Higher Education Institutions (PAPD) This work was supported by Science and Technology Project of Huangshan City, Grant number: 2017KN-06. Nanjing Forestry University Research and Development project (JZ20190011). Priority Academic Program Development of Jiangsu Higher Education Institutions (PAPD). The funders had no role in study design, data collection and analysis, decision to publish, or preparation of the manuscript.

==============================
Dendrobium officinale is a medicinal herbal plant with important health care value and high demand. Due to its slow growth and scarcity in nature, its yield depends on intensified cultivation while biotic and abiotic stresses were important factors that causes production loss. Orchidaceae can form association with rhizoctonias collectively, and studies have found that some orchids showed a high level of strain-species specificity to orchid mycorrhizal fungi (OMF), yet the specificity of OMF on D. officinale needs to explored. In this study, the effects on D. officinale of four OMF isolated from Cymbidium were tested. The obviously higher mass yield of the treated plants in medium and pots indicated the growth promotion effect of the fungi. Furthermore, an abiotic stress test indicated stronger drought tolerance among the treated plants. For the biotic stress test, two root rot pathogens, Fusarium solani and Fusarium graminearum , were isolated and identified from root rot of D. officinale. In an in vitro inhibition test, the four OMF could resist the growth of these pathogens. In vivo studies showed that these four OMF could improve the survival rate and fresh weight and decrease the root rot rate of pathogen-inoculated seedlings. The four OMF namely; Hyphomycete sp., Umbelopsis sp., Ceratorhiza sp. and Ceratorhiza sp. are compatible strains for improving the growth rate of D. officinale by increasing its environmental stress tolerance, providing an effective way to supply resources through artificial reproduction.

Introduction

Dendrobium officinale Kimura et Migo belongs to Orchidaceae and has high medicinal value. It has been widely used in Asian countries for hundreds of years to enhance immunity, provide antithrombotic and antineoplastic effects, and delay ageing (Zhao, Xu & Hua, 2017; Liu et al., 2011; Xiang et al., 2013; Ng et al., 2012). Due to the rising demand for this herb, wild D. officinale is severely exploited and will become extinct. Currently, massive amounts of D. officinale are rapidly propagated under intensified cultivation to overcome this problem (Tang et al., 2017). However, issues in artificial cultivation have also emerged, such as low growth rate, poor adaptability and plant disease. Notably, susceptibility to withering and root and stem rot mainly caused by fungal disease has resulted in serious economic losses in the cultivation of this plant (Lin et al., 2011; Zhou et al., 2017).

Mycorrhizal fungi are a group of fungi that live in healthy plants at a certain or all life stages (Otero, Ackerman & Bayman, 2002; Tejesvi et al., 2007). Mycorrhizal fungi can form a mutually beneficial relationship with host plants. Orchid mycorrhizal fungi (OMF) are present in all orchid species and play an important role in the growth of orchid species by establishing obligate relationships conducive to gain carbon, nutrients and water, especially during early life stages when orchids lack sufficient nutrient reserves in seeds (Dearnaley, Martos & Selosse, 2012; Dressler & Rasmussen, 1996). While orchid acquires nutrients by mycoheterotrophy, the root function of the host can be strengthened by the OMF to increase the absorbability of mineral elements, especially phosphorus (Bidartondo, 2005). OMF can also stimulate the growth and development of host plants, improving the host’s resistance to stress (Aly, Debbab & Proksch, 2011; Aly et al., 2010). Applying suitable OMF could be advantageous for orchid conservation and reproduction, e.g., propagation, ex situ seeding or germination. However, the acceptability of and preference for OMF by the host is an important factor to consider before application (McCormick et al., 2006; Girlanda et al., 2011; Xing et al., 2013). Mycorrhizal fungi may have different effects on different hosts, which may depend on the process of recognition between the host and mycorrhizal fungus as well as compliance with symbiosis (Klironomos, 2003; Raju et al., 1990; Kosuta et al., 2003). Suitable mycorrhizal fungi are needed for seed germination and seedling growth of D. officinale under natural conditions (Jacquemyn, Duffy & Selosse, 2017; Hossain et al., 2013). Even though adequate nutrition is provided by artificial cultivation, this plant is slow-growing and sensitive to biological and abiotic stressors. This is probably because appropriate mycorrhizal fungi are deficient in artificial cultivation conditions.

Root rot is a major disease of D. officinale that is mainly caused by fungal pathogens. It can weaken the root function of plants until the whole plant dies (Bodah, 2017). This disease can cause massive infection and death in intensive cultivation of D. officinale (Lin et al., 2011; Zhou et al., 2017). Fungal pathogens in Dendrobium have been identified, such as Fusarium sp., Pythium sp. and Ceratobasidium sp. (Lin et al., 2011; Zhou et al., 2017; Zhang et al., 2017). Chemical agents s uch as metalaxyl-M, propamocarb, and Bordeaux mixtures have been widely used to prevent and treat this disease (Sardrood & Goltapeh, 2018). However, chemical residues on the products inhibit the growth of probiotic fungi and lead to low-quality products that are harmful to sustainable agricultural development (Mahanty et al., 2017; Singh, Pandey & Singh, 2011).

In view of the above problems in intensified cultivation, four OMF (namely GDB162, GDB254, GS222, MLX102) isolated from other Cymbidium were used to promote the growth and to increased abiotic and biotic stress tolerance of D. officinale because these strains reportedly showed growth promotion effect on Cymbidum such as C. hybrid, C. goeringii, C. mastersii in former studies (Guo et al., 2012; Wu et al., 2013; Dong, 2008). On the other side, the pathogenic fungi isolated and identified due to the root disease is still a problem in intensified cultivation of D. officinale (Lin et al., 2011; Zhou et al., 2017). Further, these OMF were used to prevent and control root rot D. officinale. Our findings have great significance in increasing and reducing the morbidity of D. officinale under intensive cultivation.

Materials & Methods

Plant materials, fungi and medium

The four mycorrhizal fungi are GDB162 (Hyphomycete sp.) (Dong, 2008), GDB254 (Umbelopsis sp.) (Dong, 2008), GS222 (Ceratorhiza sp.) (Dong, 2008; Jin et al., 2007) originating from Cymbidium faberi Rolfe in Anhui Province, China, and MLX102 (Ceratorhiza sp.) (Guo et al., 2012; Wu et al., 2013; Dong, 2008), which was isolated from Cymbidium sinense (Jackson ex Andrews) Willd in Yunnan Province, China. D. officinale with root disease was collected from a nursery in Wujiang city, Jiangsu Province, China. Fungi were incubated on PSA medium containing 20 g sucrose, 20 g agar, 200 g potato and 1,000 mL water. Mycorrhizal fungal infection of D. officinale was facilitated by DE symbiotic medium (Dijk & Eck, 1995) containing K2SO4 1.0 mg/L, FeSO4 100 mg/L, MnCl2 3.3 mg/L, NaMoO4 1.0 mg/L, MgSO4 0.5 mg/L, KH2PO4 0.4 mg/L, H3BO4 25 mg/L, ZnSO4 2.8 mg/L, yeast extract 10 g/L, 9 g/L soluble starch and 4.5 g/L agar powder (pH 6.0). MurashigeSkoog1962 (MS) medium was used as a medium for the growth of D. officinale seedlings (Murashige & Skoog, 1962). The rooting medium was MS medium plus 1 mg/L NAA (1-naphthylacetic acid).

The plant effects of mycorrhizal fungi on D. officinale

D. officinale PLBs were cultured in rooting medium. After that, three seedlings with similar growth conditions (approximately 1.5 g of each) were transferred to solid DE symbiotic medium and cultured at 25 °C for two days to verify the absence of contamination. Three five mm diameter blocks of agar containing fungi were evenly inoculated around D. officinale seedlings. For the testing of tissue culture seedlings, the seedlings were cultured at 25 °C and 12 h daily light for 60 days. Then, fresh weight, chlorophyll content, stem polysaccharide levels, and root acid phosphatase activity (ACP) were measured and analyzed. For the testing of pot seedlings, the seedlings were transplanted to the pots after 15 days of OMF inoculation and were grown in a closed green house with natural light at 25 °C for 5 months. Then, the survival rate, stem length, stem diameter and leaf number were measured and analyzed. The experiment was performed in thirty replicates.

The effect of mycorrhizal fungi on plant drought resistance

D. officinale seedlings in pot with similar growth condition and weight were selected for testing of drought resistant. The four leaves (about five mm around wide) of D. officinale infected by mycorrhizal fungi were placed around the root of the seedling for mycorrhizal fungi inoculation and the CK group consist of leaves without mycorrhizal fungi incubation (Gilbertson, 1988; Munro et al., 1999). The mycorrhizal fungi infected the root of the plant and adequate water was supplied. Then, three experimental groups were set up. The group I (CK) was adequately water was supplied every week for the whole time. The group II was subjected to limited water supply every week, and the group III was subjected to adequate water was supplied only once at the beginning. After that, the survival rate of the plant of group III was recorded at 15, 30, 60 days and the MDA content of group I and II were measured and analyzed at 60 days. Each group had 10 replicate. Malonaldehyde (MDA) content was determined with the method of thibabituric acid (TBA) by using an MDA assay kit (cat: A003-1-2; Nanjing Jiancheng Bioengineering Institute, Nanjing, China) according to the manufacturer’s instructions and determined at 532 nm. The experiments were performed in triplicate.

Determination of chlorophyll, and phosphatase activity

Chlorophyll a and b were determined as described previously (Şükran, Güneş & Sivaci, 1998). Fragmented fresh leaves (0.25 g) were transferred into a mortar, and then 2.5 ml of acetone was added. The mixture was ground to a homogenate, and 2.5 ml of 80% acetone was added. The samples were centrifuged at 12,000× g for 10 min. The volume of the supernatant was adjusted to 10 ml with 80% acetone. Then, 0.5 ml chlorophyll extract was diluted with 2 ml of 80% acetone, and the solution was transferred to a cuvette. The 80% acetone solution was used as a control. The absorbance was measured by a spectrophotometer at 645 and 663 nm to determine the chlorophyll a and b contents, respectively.

Acid phosphatases activity was assayed using ρ-nitrophenol. Fresh root (0.5 g) was transferred to a mortar and pestle at 0–4 °C in 50 mM sodium acetate buffer (pH 5.5). The sample were centrifuged at 12,000 g for 15 min and the supernatant collected. ACP was determined using ρ-nitrophenol phosphate as substrate and measuring the amount of ρ-nitrophenol produced. Activity was quantified by comparing the absorption at 410 nm to a standard curve of diluted ρ-nitrophenol solutions and NaOH. One unit of phosphatase is equivalent to the amount of enzyme producing 1 µM of product per min under assay conditions.

Pathogen isolation and identification

The pathogens were isolated from rotted roots of D. officinale. The rotted roots were soaked in 70% ethanol for 5-10 s, then in 0.1% mercury bichloride for 90 s and washed with sterilized water three times. The treated roots were incubated on PSA medium at 26 °C for 3-5 days. All suspected mycelium tips of pathogens were transferred to new PDA medium. The following three primer pairs were used to amplify the fungal rRNA internal transcribed spacer (ITS), the second largest RNA polymerase subunit (RPB2) and large fragment of ribosomal (LSU) fragments: ITSF (5′- TCCGTAGGTGAACCTGCGG-3′)/ITSR (5′- TCCTCCGCTTATTGATATGC-3′) (White et al., 1990), RPB2-5F2(5′- GGGGWGAYCAGAAGAAGGC)/7cR (5′- CCCATRGCTTGYTTRCCCAT) (Anonymous, 2007) and LSUF LSUF (5′- ATCCTGAGGGAAACTTC-3′)/LSUR (5′- GTACCCGCTGAACTTAAGC-3′) (Sung et al., 2007).

Pathogenicity test

A pathogenicity test was performed on D. officinale by following Koch’s postulates with tissue culture seedlings and plants in pots. The seedlings were incubated on DE medium for 2 weeks at 26 °C with a 12-h dark/light cycle. The pathogenic fungi were evenly inoculated around seedlings for 30 days of co-incubation. The plants in pots containing dried bark and peat were grown at a ratio of 2:5 at 26 °C for 15 days. The fungi were incubated in Czapek medium with 120 rpm shaking and 25 °C for 4 days. Then, the conidia suspensions (106 CFU/ml) were created and inoculated into the rhizosphere of plants in pots. The plants were incubated at 26 °C and 65% relative humidity for 30 days, and the root rot rate and death rate were recorded. The root rot severity was rated on a scale of 1 (root no wilting, white and plump) to 5 (plant death, root complete yellow and shrinking) (Meyer & Hausbeck, 2013). The experiments were performed in triplicate.

Antagonism studies in vitro

The mycorrhizal fungi and pathogens were inoculated on PSA medium (2 cm away from the edge in a nine cm petri dish) at an interval of 5 cm (Dubey et al., 2013; Dubey, Dan & Karlsson, 2014). The inoculated plate was incubated at 20 °C for 1 week, and the colony diameters of both fungi were measured every 24 h. The experiments were performed in triplicate.

The effect of mycorrhizal fungi on pathogens

Three 5 mm diameter blocks of agar containing OMF were evenly inoculated around D. officinale seedlings respectively. Then, the seedlings were incubated at 25 °C under 12 h daily light for 15 days. After that, the co-incubated seedlings were transferred to pots in a greenhouse, and the seedlings were inoculated with 1 mL of the conidia suspensions (106 CFU/ml) of the pathogenic fungi. The survival rate, total root rot and flesh weight were recorded after 30 days. Ten replicates of the experiment were performed.

Statistical analysis

Statistical analyses were performed using SPSS 17.0. One-way analysis of variance (ANOVA) and Tukey’s test were used to detect the differences in seedling weight, polysaccharide rate, chlorophyll content, acid phosphatase activity, MDA content, colony diameter and root rot rate, survival rate.

Results

The promotion of growth of tissue-cultured D. officinale by mycorrhizal fungi

Four mycorrhizal fungi were inoculated to test their growth-promoting effect on tissue-cultured D. officinale. The results showed that these four strains promoted the growth of D. officinale for 60 days. The seedlings demonstrated better growth, with bright green shoots (Fig. 1), higher fresh seeding weight (Fig. 2A), higher chlorophyll contents (Fig. 2B), and higher levels of polysaccharide in the stems (Fig. 2C) compared with those in the control group (CK). The differences between the stimulatory effects of GS222, GDB254, and MLX102 on the pigment contents were not significant. MLX102 had the strongest promoting effect on chlorophyll a levels, which demonstrated a 1.85-fold increase compared to that in the CK. GDB254 was the most effective stimulator of chlorophyll b (2.72-fold) and total chlorophyll (1.94-fold) contents compared to those in the CK. GDB254-treated plants had a greater increase in weight. The acid phosphatase activity (Fig. 2D) indicated that the phosphorus absorption of inoculated D. officinale roots was weaker than that in the CK roots.

Figure 1 Mycorrhizal fungi promote the growth of tissue culture seedlings of Dendrobium officinale.

GDB254, MLX102, GS222, and GDB162 were inoculated into D. officinale seedlings for 60 days and were associated with increased weight acquisition compared to that in the control group (CK).

Figure 2 Mycorrhizal fungi promote the growth of tissue culture seedlings of Dendrobium officinale.

The growth-promoting effects of GDB254, MLX102, GS222, and GDB162 on D. officinale tissue culture seedlings after 60 days. (A), (B), (C), and (D) show the fresh weight, chlorophyll content, stem polysaccharide levels, and root acid phosphatase activity of D. officinale, respectively; the values of the parameters increased after symbiotic growth with different mycorrhizal fungi. Letters over the bars indicate significant differences at the 5% level.

The promotion of growth of D. officinale in pots by mycorrhizal fungi

To study the promoting effect of these mycorrhizal fungi on D. officinale, four fungi were inoculated into cultivated seedlings in pots. The results showed that all the mycorrhizal fungi had a growth-promoting effect on the seedlings after for 5 months. The seedling survival rate after transplantation from the medium to the substrate in the GDB254, MLX102, GS222, and GDB162 groups was increased by approximately 16%, 13%, 10%, and 13%, respectively, compared with that in the CK group (Fig. 3A). The stem diameter, stem height, and leaf number of the seedlings inoculated with mycorrhizal fungi (Figs. 3B, 3C, 3D) were significantly increased. GDB162 had the most efficient growth-promoting effect of all mycorrhizal fungi and induced an increase in seedling thickness, height, and leaf number by approximately 118%, 54%, and 58%, respectively.

Figure 3 Growth-promoting effects of mycorrhizal fungi on Dendrobium officinale cultivated in pots.

(A) Effects of mycorrhizal fungi on the survival rate after transplantation from the culture bottles to flower pots. (B, C, and D) Growth-promoting effects of various mycorrhizal fungi on (B) the stem diameter, (C) stem length, and (D) leaf number of the plants grown in pots. Different letters over the bars indicate significant differences at the 5% level.

Drought stress resistance of D. officinale due to mycorrhizal fungi

To evaluate the effect of these fungi on the drought resistance of D. officinale, the survival rate and MDA (malondialdehyde) content were measured under various drought conditions. The results showed that the survival rate of D. officinale was significantly improved in the 15, 30, 60 days of drought conditions in the presence of mycorrhizal fungi (Fig. 4A). In the GDB162, GS222, and GDB254 tested groups, the survival rate of D. officinale at various drought periods was improved by more than 40% compared with the CK group. On the other hand, the MDA content in mycorrhizal fungi inoculation groups was steadier and maintained at a low level in the drought period, while the MDA content of the CK group during drought is increased to twice times that of the normal growth environment. This phenomenon was more obvious in the MLX102 and GDB254 groups than in the other groups (Fig. 4B). In addition, most of seedlings in group III were dead, hence, MDA content analysis was not conducted for this group. The results indicated that the four mycorrhizal fungi could promote drought resistance in D. officinale.

Figure 4 The effect of mycorrhizal fungi on the drought resistance of D. officinale.

(A) The effect of mycorrhizal fungi on the survival rate under drought conditions. (B) The effect of mycorrhizal fungi on the MDA contents of D. officinale under various drought conditions.

Isolation and identification of pathogenic fungi

Two fungi, TS1 and TS2, were isolated from D. officinale with root rot and showed pathogenicity (Fig. 5). The sequence data of TS1 showed 100% homology to Fusarium solani (GenBank accessions ITS-MT638068.1 (449/449 bp), RPB2-MK606410.1 (861/861 bp), LSU-MT533257.1 (726/726 bp), and the pathogen was identified as Fusarium solani. Sequence data of TS2 showed homology to Fusarium graminearum (GenBank accessions ITS-KU254606.1 (789/789 bp), RPB2-LT222053.1 (970/976 bp), LSU-MH877271.1 (857/857 bp), and the pathogen was identified as Fusarium graminearum.

Figure 5 Morphology and microstructure of pathogenic fungi.

Morphology of TS1 (A) and TS2 (D) strains grown on PSA. (B, C) Microscopic morphology of (E, F) Microscopic morphology of TS2. Colonies of TS1 were formed after 3 days of growth on PSA medium at 28 °C.

Pathogenicity test of the pathogens

TS1 and TS2 show pathogenic effects on D. officinale in medium (Fig. S1) and pot conditions. The pathogen-incubated groups showed root rot symptoms after two weeks (Fig. 6), while the roots of the control group were healthy and strong. TS1 was strongly pathogenic; most seedling’s root turned yellow and withered while the survival rate of the TS2 group was 55%, and the roots turned yellow and black (Figs. 6B, 6C).

Figure 6 TS1 and TS2 strains all showed pathogenicity to Dendrobium officinale.

(A) The control group, which was not inoculated with pathogenic fungus. TS1 (B) and TS2 (C) were incubated with D. officinale seedlings. The solid arrows point to the rot root and hollow arrow point to healthy root.

Competitiveness of the four mycorrhizal fungi with pathogenic fungi

To test the interaction of four mycorrhizal fungi and pathogens, these strains were used to confront these pathogens in vitro. All the pathogenic fungi grew with mycorrhizal fungi on the medium, but the pathogens had a faster growth rate. The growth of TS1 was significantly delayed under the effect of mycorrhizal fungi (Figs. 7A–7D). The growth curve indicated that the growth rate of TS1 was obviously slowed down in 3–5 days when it came in close proximity to the mycorrhizal fungus. The inhibition of mycorrhizal fungi on TS2 was not as obvious as that on TS1, in which colonies touched each other (Figs. 7E–7H).

Figure 7 The competitive effect of mycorrhizal fungi with pathogenic fungi.

The mycorrhizal fungi (GDB254, GS222 MLX 102, and GDP162) competed with the pathogenic fungi (TS1 and TS2) by means of orthogonal experimental design, and the growth curves of the pathogenic fungi were plotted and analysed. (A–D) The mycorrhizal fungus competing with TS1 and (E–H) the mycorrhizal fungus competing with TS2.

Mycorrhizal fungi improve root rot resistance of D. officinale

The competitive effect between the mycorrhizal fungi and pathogens in vitro did not completely support the biocontrol potential of the mycorrhizal fungi in the hosts. Thus, we evaluated the potential of mycorrhizal fungi as biocontrol agents against pathogens in D. officinale on DE medium and in pots in greenhouses via interactions among hosts, mycorrhizal fungi, and pathogens together. The survival rate of D. officinale indicated that four mycorrhizal fungi played roles in improving the survival ability of tissue culture seedlings (Fig. 8). The highest survival rate (∼50% in TS1 and ∼80% in TS2) indicated that MLX102 supplied stronger disease resistance for D. officinale. The mycorrhizal fungus increased the survival rate to a limited extent (0∼20%) due to the weaker lethality of TS2 (Fig. 8A). The root rot rate and fresh weight of D. officinale in the greenhouse both indicated improved disease resistance to pathogenic fungal infection. The root rot rate of the CK groups (>90%, Fig. 8B) indicated that both pathogenic fungi were highly virulent. MLX102 inhibited the root rot rate caused by TS1 and TS2 with an efficiency of approximately 40%∼50% (Fig. 8). GS222 seemed unable to inhibit root rot caused by TS1, but it had a certain effect on TS2, with an inhibition rate of approximately 25%. The other two mycorrhizal fungi had more moderate inhibition effects, with inhibition efficiencies between those of MLX102 and GS222. All the mycorrhizal fungi led to an increase in fresh weight to some extent. Although the reduction in root rot rate did not correspond to an increase in fresh weight, all the different mycorrhizal fungi have the effect of reducing the root rot rate and increasing the fresh weight (Fig. 8C).

Figure 8 Mycorrhizal fungi increase the resistance of Dendrobium officinale.

Mycorrhizal fungi interacting with D. officinale could increase its disease resistance and thus improve its survival rate on DE medium (A), inhibit root rot (B) and increase fresh weight yield (C) under disease stress.

Discussion

Mycorrhizal fungi are important resources for D. officinale growth promotion and intensify their tolerance to the environment (Chen, Wang & Guo, 2012), with promising agricultural applications (Vega, 2018; Shahzad et al., 2018). But the degree of specificity of most orchids to their mycorrhizal associates remains unknown. In this study, we have shown that mycorrhizal fungi of Cymbidium could promote the growth of D. officinale tissue culture seedlings and pot cultivation seedlings. Interestingly, mycorrhizal fungi improved the adaptation of D. officinale to biotic stress and abiotic stress resistance. It is important to note that the mycorrhizae from Cymbidium are compatible with D. officinale. Literatures have shown that mycorrhizal fungi can promote the growth of orchids as well as stimulation of germination (Brundrett, 2007; Jacquemyn et al., 2015; Zi et al., 2014). However, few studies employed orchid-mycorrhizal fungi to promote the growth of D. officinale. Moreover, we also found and identified the root rot pathogen (soil-borne) of D. officinales and mycorrhizal fungi could increase such biotic stress tolerance. This finding will improve our understanding of the promotion effects of OMF on D. officinale.

According to reports, OMF have contributed to the growth, development, nutrient uptake, and resistance to pathogenic infection of different plant species. These four fungi have been proven that OMF could promote growth of orchids such as C. hybrid, C. goeringii, C. mastersii in former studies (Guo et al., 2012; Wu et al., 2013; Dong, 2008). In this study, we were able to prove that OMF significantly contributed to both physiological and morphological parameters of D. officinales expressing different growth effect on D. officinale. The stem length, diameter and polysaccharide levels all increased, indicating that the quality of the seedlings was improved.

The biochemical index quantificationally also demonstrated the promotion effect of these fungi. Our results show that all four mycorrhizal fungi could increase the weight as well as improve the content of photopigments (chlorophyll a, chlorophyll b and total chlorophyll), on average. The increase of chlorophyll in the symbiotic plant probably resulted in higher photosynthetic rates and thus improved plant biomass. The chlorophyll b content of D. officinale in the medium was especially obvious. As an index of blue and violet light absorption, the high chlorophyll b content indicated that mycorrhizal fungi could promote low light capacity utilization (Ohashi-Kaneko et al., 2010). This increase of chlorophyll content by mycorrhizal fungi inoculation may be due to an increase in stomatal conductance, photosynthesis, transpiration (Kirschner, 2013; Rajasekaran et al., 2006).

AcPase activity describes the phosphorus usage efficiency of plants. The decreasing phosphatase efficiency in the mycorrhizal fungi coculture groups indicated that adequate phosphorus was supplied. The OMF supplied the orchid with phosphorus (P) demonstrating that the OMF was involved in P uptake and transfer in former studies (Cameron et al., 2007; Alexander & Hadley, 1984), which could provide additional phosphorus to P acquired by the root of host plant. Similar studies of OMF proves that P absorption and transfer abilities could provide phosphorus for host plant usage (Plassard, Becquer & Garcia, 2019; Siti et al., 2013). Furthermore, difference in ACPase activity could be attributed to the metabolism of plant host during interaction with OMF where the cell of OMF is digested in cavity and the activity occurs with some level of variation (Jin et al., 2007; Liu, 1982). In addition, the AcPase activity of root could be different in root after different mycorrhizal fungi species invade the host plant (Conn & Dighton, 2000).

OMF could enhance the absorption ability of the root system as well as the hydraulic conductivity of plants, which is associated with drought tolerance (Zaidi et al., 2009; Smith et al., 2010). The increased survival rate and decreased MDA content under drought conditions indicated that OMF increased the drought tolerance of D. officinale. The potential of the mycorrhizal fungi to confer drought resistance to host plants is important for plant adaptation to the environment (Singh, Gill & Tuteja, 2011; Rapparini & Peñuelas, 2014).

Root rot caused by pathogenic fungi has led to great economic losses in the production of D. officinal e and orchids (Bodah, 2017; Wang et al., 2014; Latiffah et al., 2010). Two pathogenic fungi were confirmed: TS1 and TS2 were Fusarium solani and Fusarium graminearum, respectively, while the severe root rot in crops and fruits could be caused by these fungi (Bodah, 2017; Wang et al., 2014; Mesterhazy, Lemmens & Reid, 2012; Nemec & Zablotowicz, 1981; Bohra & Mathur, 2004; Lee, Zhang & Waalwijk, 2015; Kikot, Hours & Alconada, 2009). Fusarium species are one of the most common soil-borne pathogens for plants and are causal agents of orchid root rot (Swett & Uchida, 2015; Latiffah et al., 2008). These pathogens are major problems for commercial orchid production (Swett & Uchida, 2015; Srivastava, 2014).

These pathogens also caused root rot in D. officinale in this study. TS1 is a highly pathogenic fungus that causes a high lethal ratio under sterile medium conditions. However, the seedling roots tended to rot in the pot after being inoculated with this pathogen. This pathogen is relatively more likely to cause a high rate of root rot in greenhouses. This difference may be because the DE medium is also suitable for fungal growth. It provides fungi with adequate nutrients, which means that it helps pathogen growth and exerts virulence (Lu et al., 2013; Fernandez, Marroquin-Guzman & Wilson, 2014). The four mycorrhizal fungi in this study exhibited resistance to pathogens on plates and pots, which was mainly manifested both in the increased survival rate of seedlings and in reduced root rot of the plants. They exhibited different inhibition zone diameters in the plate competition test, which may be caused by antibiotics and lytic enzymes produced by the fungi that has reported in other studies of mycorrhizal fungi (Wanderley et al., 2017; Krywolap, Grand & Jr, 2011; Santoro & Casida, 1962). These effects of the mycorrhizal fungi of D. officinale are essential to the resistance of root rot pathogenic fungi in production.

Biological control has advantages, such as relatively few side effects, less pollution and longer effective times, and is a major trend in the development of agriculture and forestry today (Vega, 2018; O’Brien, 2017). This is an effective biocontrol method that is widely used in farming (Toghueo et al., 2016; Egamberdieva et al., 2017; Jaber & Ownley, 2018), and it is also suitable for use in the production of D. officinale today. Mycorrhizal fungi could be important biocontrol resources for competing with and inhibiting the reproduction of the pathogens by producing antibiotic or antifungal compounds and stimulating plant defence responses (Dell, 2002; Gao, Dai & Liu, 2010). Abiotic stress could weaken plant defences and enhance pathogen infection probability (Mittler & Blumwald, 2010). It has been suggested that the combination of abiotic and biotic stress could cause increased serious disease (MacDonald, 1982; Pandey, Sinha & Senthil-Kumar, 2015). Mycorrhizal fungi could enhance resistance to both stresses, indicating that these fungi are useful resources for the growth promotion of D. officinale.

Conclusions

Together, the mycorrhizal fungi in the culture medium or the cultivated substrate could promote the production of D. officinale by increasing resistance to biological and abiotic stress. Furthermore, we isolated and characterized two fungal root pathogens of D. officinale. Inhibition of fungal plant pathogens by mycorrhizal fungi isolated from different Cymbidium was a suitable means to control the disease caused by Fusarium. This finding is important for growth promotion, root rot disease prevention and control in D. officinale production.

Supplemental Information

Supplemental Information 1 The effect of TS1 and TS2 on D. officinale in medium

Click here for additional data file.

Supplemental Information 2 Raw data for Figs. 2–4 and Figs. 7–8

Click here for additional data file.

Supplemental Information 3 Sequences of ITS, RPB, LSU of molecular identifications

Click here for additional data file.

Additional Information and Declarations

Competing Interests

Author Contributions

DNA Deposition

Data Availability

Ping Sun is employed by Jiangsu Aosaikang Pharmaceutical Co., Ltd.

Yulong Li conceived and designed the experiments, performed the experiments, prepared figures and/or tables, authored or reviewed drafts of the paper, and approved the final draft.

Zhihua Kang and Xia Zhang performed the experiments, prepared figures and/or tables, and approved the final draft.

Ping Sun analyzed the data, prepared figures and/or tables, and approved the final draft.

Xiaohui Jiang analyzed the data, prepared figures and/or tables, authored or reviewed drafts of the paper, and approved the final draft.

Zhengmin Han conceived and designed the experiments, analyzed the data, authored or reviewed drafts of the paper, and approved the final draft.

The following information was supplied regarding the deposition of DNA sequences:

The TS1 sequences are available at GenBank: ITS-MW800179, RPB2-MW816630, LSU-MW811461.

The TS2 sequences are available at GenBank: ITS- MW800198, RPB2- MW816631, LSU- MW800199.

The following information was supplied regarding data availability:

The raw data are available in the Supplemental File.

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
