# Peer review of "The mycorrhizal fungi of Cymbidium promote the growth of Dendrobiumofficinale by increasing environmental stress tolerance"

_PeerJ, doi:10.7717/peerj.12555_

## Round 0.1 · original submission · Major Revisions

In view of the criticisms of the reviewers found at the bottom of this letter and my own assessment, a revision of your manuscript is required for publication in PeerJ. For revision, I expect the authors to read the reviewers' comments thoroughly and revise the manuscript and add the data to address all concerns raised by reviewers.

Reviewer 1 ·

Basic reporting

In their study, Li et al. used four mycorrhizal fungi isolated from Cymbidium to inoculate with Dendrobium officinale under biotic and abiotic stresses.The results showed that the four Cymbidium mycorrhizal fungi are compatible strains for improving the growth rate of D. officinale by increasing its environmental stress tolerance, providing an effective way to supply resources through artificial reproduction.The nature of the study is technical as it provides information about mycorrhizal fungi can improving the growth rate of D. officinale by increasing its environmental stress tolerance on the basis of the results.

Experimental design

After reading your manuscript, it was not really clear for me, what is your main obtained result. For me the manuscript is quite confusing, especially “four mycorrhizal fungi isolated from Cymbidium”.

Validity of the findings

First I strongly suggest you to search in available literature for information. Than I recommend to add some nice pictures showing the morphological characteristics of studied plants and fungi. The English language should be improved and I suggest that give us more clearly information in section of “Material and methods”

Additional comments

Below I state some comments/suggestions that could improve their study.
1. L43-L46 We suggests that you may provide more relevant information about Orchidaceae mycorrhizal fungi (OMF). Orchidaceae mycorrhizal fungi are a large group with specific genera and species.
2. L74-75, In this paper, authors showed the hosts of strains GDB162, GDB254, GS222 and MLX102, but not give more information about these four strains. You have to identify these strains, and give the corresponding information.
3. L90, Please tell us why choose “60 days” for data determination?
4. L91 , What do you mean” other parameters” ? Please write in detail.
5. L94 What do you mean “4 leaves of D. officinale”?
6. L97 Why do you know “infected the root of the plant (about 15 days)”? I suggest that you need to provide tissue sections of the root to determine the colonization time.
7. L98-100, What are the test conditions? Pot?or MS medium?What do you mean”The first” and ”The second” ? I think it should be “Group I” and “Group II”.
8. L132, How do you measure “symptoms and death rate”? I suggest that you need to provide references.
9. L148, I suggest that you should add the “method of acid phosphatase activity “ in the section of “material and method”.
10. Figure 1, I recommend to add some nice pictures.
11. L165-L174, in figure3, You did not mention “5 month” and “bud number” in the section of “material and method” , please give us more information.
12. L179, What do you mean” in the early, middle, and late periods of drought conditions”? In supplements, 15,30,60 days, Does it mean the same thing? I recommend to describe clearly.
13. Please add the rulers in figure 5A and D.
14. L189-L200, I recommend to use shorter but more exact sentences, or you can delete it, as it adds nothing to the study.
15. L209, What do you mean “most seedlings withered”? Please give the statistics. Please add the rulers in figure 6.
16. L213, “four Rhizoctonia spp”? Four strains used in this study belonged to genus “Rhizoctonia”? or other four Rhizoctonia spp.?
17. L248-255, I suggest that you need to use shorter but more exact sentences to describe the pathogens information. We want to know more details about the function of mycorrhizal fungi, instead of pathogens.
18. L265 “mainly caused by antibiotics and lytic enzymes produced by the fungi on the plates”, Please provide more references.
19. L274-L293, Many literatures have shown that mycorrhizal fungi can promote the growth of orchids. Please explain the similarities and differences between your research and others' research.
20. L284, “Mycorrhizal fungi that have dissolving abilities should provide additional phosphorus for usage”, please provide references. There are many types of mycorrhizal fungi. The interaction mechanisms between plants and fungi are different. What kinds of mycorrhizal fungi do you mean?

Reviewer 2 ·

Basic reporting

Language is fluent. but background of the mycorrhizal fungi GDB162, GDB254, GS222 and MLX102 are not enough!

Experimental design

1) In materials & methods,the author have not designed the mrcorrhizal fungi proof for GDB162, GDB254, GS222.
2) In materials & methods,the author designed the content "Determination of chlorophyll and MDA contents",but no results were given. the author only mentioned it in discussion.

Validity of the findings

no comment

Additional comments

74-75line: Because GDB162, GDB254, and GS222 are mycorrhizal fungi, you should give the reference about GDB162, GDB254, and GS222. If you first report you should draw a cluster tree diagram and give the proof of mycorrhiza.
102line: I have not found the results about the content “Determination of chlorophyll and MDA contents”
140-141 line: what is the concentration of mycorrhizal fungus to inoculate into D. officinale?
142 line: “the seedlings were inoculated with pathogenic fungi”. The fungal spores or colony, which one was used to inoculate? What is the concentration of pathogenic fungi?
207-211 line: please give the picture “most seedlings withered”;on the other hand, please mark the diseased roots with arrows.
212 You mentioned “Mycorrhizal fungi are competitive with pathogenic fungi”, can GDB162, GDB254, GS222 and MLX102 colonize the D. officinale? But you have not given the proof.
221-239 line: you give the result “Mycorrhizal fungi improve root rot resistance of D. officinale”by chart. Can you give the picture for pot experiment?

---

## Round 0.2 · Minor Revisions

The authors seem to address all concerns raised by reviewers and revised the manuscript according to the comments from them. However, one reviewer still has comments on the revised manuscript. Hence, I would like to wait for a proper revision before acceptance for publication.

Reviewer 2 ·

Basic reporting

This manuscript describes that four mycorrhizal fungi of Cymbidium can promote the growth of plants and increase environmental stress tolerance on Dendroium offcinale. One regrettable thing is that whether the four mycorrhizal fungi can colonized on roots of D. offcinale had not been directly varified. However, the study can guide agronomic operation to enhance production of D. offcinale to some extent.

Experimental design

no comments

Validity of the findings

no comments

Additional comments

1. In materials & methods, the information of GDB162, GDB254, GS222, MX102 has already provided on the manuscript. But I want to know that whether has some references reported for these four strains? If yes, please provide?
2. line 105-106: why did the experiment use leaves to inoculate mycorrhizal fungi? Please provide references if possible. Moreover, on the part: the effect of mycorrhizal fungi on pathogens (line163), you used 5mm diameter blocks of agar containing fungi.
3. line 109-111: there are three group on here,but only two group (adequate and limited water) were showed in figure 4. why?
4. line157-161: Whether this protocol has been reported previously, and if available, please provide the literature.
5. line 196-198: According to figuer3B-D, the fungus with the best growth-promoting effect is GDB162, not GDB254.
6. line 201-203: There are three group in material & method, which experiment groups are described by this result ?
7. line219-220: Pathogenicity tests were conducted on medium and pot conditions. However, figure 6 showed only plants in pot, please provide information on medium. In addtion, the material & method about this test should be revised clearly.
8. line 235-238: Likely Pahtogenicity, this experiment were conducted on medium and pot conditions, but figure 8 seemed to show only plants in mediumt, please provide more information on another condition. In addtion, the material & method about this test should be revised clearly.
9. The materials & methods section of the manuscript should be revised for clearer.
10. Another comment, please see the attachment.

Annotated reviews are not available for download in order to protect the identity of reviewers who chose to remain anonymous.

---

## Round 0.3 · accepted · Accept

I confirmed the authors have addressed all concerns raised by the reviewer. Hence, I think that the current version of the manuscript deserves publication in PeerJ.